# *Exserohilum turcicum* (Passerini) Leonard and Suggs: Race Population Distribution in Bihar, India

**DOI:** 10.3390/bioengineering10010007

**Published:** 2022-12-21

**Authors:** Ram Niwas, Md Arshad Anwer, Tushar Ranjan, Abhijeet Ghatak, Khushbu Jain, Jitesh Kumar, Aditya Bharti, Neha Kumari, Jitendra Nath Srivastava

**Affiliations:** 1Department of Plant Pathology, Bihar Agricultural University, Sabour 813210, Bhagalpur, India; 2Department of Molecular Biology and Genetic Engineering, Bihar Agricultural University, Sabour 813210, Bhagalpur, India

**Keywords:** NCLB, TLB, *Exserohilum turcicum*, *Setosphaeria turcica*, differential lines, race, Bihar

## Abstract

Northern corn leaf blight (NCLB) of maize, caused by *Exserohilum turcicum* (Pass.) Leonard and Suggs., is an important foliar disease common across maize-producing areas of the world, including Bihar, India. In this study, virulence and distribution of races were observed against *Ht*-resistant genes and also identified the *E. turcicum* race population distribution in Bihar. For that, 45 *E. turcicum* isolates were collected from maize fields in Bhagalpur, Begusarai, Khagaria, Katihar and Samastipur districts between 2020 and 2022. These isolates were screened on maize differential lines containing *Ht1, Ht2, Ht3* and *HtN1* resistance genes. Five different physiological races were observed based on the symptoms response of the differential maize lines. These races are race 0, race 1, race 3, race 23N and race 123N. *E. turcicum* race 3 was the most prevalent race having 26.6% frequency followed by race 0 (24.4%) and race 1 (22.2%) and the least prevalent races were race 23N and 123N having 13.3% each. Varied resistance response of different isolates was observed on differential lines having different resistant genes. Despite the fact that virulence was seen against all *Ht* resistance genes, NCLB control might be increased by combining qualitative *Ht* resistance genes with quantitative resistance.

## 1. Introduction

*Exserohilum turcicum* (Passerini) Leonard and Suggs. (Perfect stage-*Setosphaeria turcica*) is the causative agent of the widespread disease known as northern corn leaf blight (NCLB) of maize (*Zea mays* L.) [1]. Green-gray, elliptical lesions that eventually turn necrotic are a symptom of NCLB disease [1]. Between 18 and 27 °C, lesions grow larger and develop more, which reduces the host’s ability to photosynthesize [2]. The primary inoculum for NCLB development is often maize residue with *E. turcicum* infection [3]. Secondary infections are caused when conidia that develop on the surface of necrotic lesions are carried to the higher canopy by wind and rain [4]. Yield losses may surpass 50% if the symptoms develop before flowering [5]. All lands where maize is grown, including India, exhibit substantial genetic diversity in the *E. turcicum* population. The best strategy to control NCLB is host plant resistance. Host plant resistance can be quantitatively controlled by multiple genes with tiny effects or qualitatively determined by single *Ht* genes that are particular to a given race. Quantitative resistance is more durable and fairly effective against all pathogen races. In contrast to susceptible NCLB symptoms, the resistance response appears as chlorotic streaks with attenuated necrotic lesions with decreased sporulation. The *Ht1* gene, also referred to as the first qualitative *Ht* resistance gene, was first discovered by Hooker in 1963 [6]. The *Ht2* and *Ht3* genes, which exhibit greater necrosis but similar resistance responses to *Ht1* [7], were first published in 1977 and 1981, respectively. In 1975, a single gene known as *HtN1* (syn. *HtN)* was found to have a longer latent period and fewer lesions [8]. The *Ht1* gene was often utilized as the primary way of NCLB management in maize breeding lines prior to the discovery of virulent *E. turcicum* populations in locations of the United States that produce maize [9]. The *Ht2*, *Ht3*, and *HtN1* genes have not been actively used in field maize breeding due to the predominance of *E. turcicum* populations virulent against these genes and heterogeneity in resistance under varied light intensities and temperatures [9]. Payak and Sharma (1985) conducted the first research on *E. turcicum* in several races in India [10]. They discovered race 1 isolates in Ludhiana, Hyderabad, Coimbatore, and Udaipur. From Godhra, Dholi, Chindwara, and Kolhapur, isolates of race 2 have been found. Race 3 isolates came from Jashipur and Jorhat, but Race 4 isolates came from Almora, Bajaura, and Nagenahalli. The four maize differentials H-4460Ht, H-4460Ht2, H-4460Ht3, and A-503HtN, which contain the multiple *Ht* genes, were found by Gowda et al. in 1993 [11]. Disease severity has increased as a result of farmers favoring maize hybrids with high yield potential over those with strong disease resistance [12]. This demonstrates that some high-yielding lines lack the quantitative resistance to manage NCLB in some places. Backcrossing for a single resistance gene can be completed in maize lines much more quickly than breeding for polygenic resistance. The qualitative *Ht* genes could increase NCLB control if combined with partial resistance, or they could provide protection while high-yield-producing lines increase their quantitative resistance. Additionally, Pataky et al. (1986) showed that qualitative resistance genes and high partial resistance were equally effective in halting the transmission and development of NCLB in the presence of a race avirulent against *Ht* genes [13]. Bihar is the leading state as far as maize production and productivity is concerned. Its production may be further increased by limiting the most serious disease, i.e., NCLB. Variations in race population diversity are challenging to explain without knowledge of the level of selection pressure present in fields and information regarding the deployment of the *Ht* resistance gene in commercial cultivars. Thus, pathogenicity and race distribution against *Ht* resistance genes were observed in this study, and it also identified the spread of the *E. turcicum* race population in Bihar, India. The knowledge might help increase the combination of qualitative and quantitative resistance genes.

## 2. Materials and Methods

### 2.1. Collection of Isolates

The *E. turcicum* collection consisted of 45 isolates obtained from maize plants from Bhagalpur, Begusarai, Khagaria, Katihar and Samastipur districts. The collection was conducted from 2020 to 2022 (Table 1). In order to stimulate sporulation, NCLB symptomatic leaves were collected and placed in humidity chambers (Plastic bags containing a damp paper towel). Conidia were collected from the leaf surface using 200–500 mL of water, and the conidial solution was then put on potato dextrose agar (PDA) (Hi-media, Mumbai, India)-coated Petri dishes with streptomycin sulphate (25 mg/litre) added to stop bacterial growth. Conidia were collected and added to a PDA after being allowed to grow. After isolation, the culture was kept in BOD at 26 °C under alternate 12-h light and 12-hdark conditions for 7 to 14 days, mycelia were removed from plates and placed in 15 mL culture tubes containing 3–5 mL PDA with 15% glycerol solution. Tubes were placed in a −80 °C freezer to be kept for a long time prior to the race-type screening of isolates.

### 2.2. Race Determination

To determine race, virulence was assessed on differential maize lines containing *Ht1*, *Ht2*, *Ht3*, or *HtN1* qualitative resistance genes or no *Ht* qualitative resistance gene. Five genotypes namely ‘6′ (no *Ht* gene), ‘33′ (*Ht1*), ‘15′ (*Ht3)*, ‘55′ (*Ht2, Ht3* and *HtN1)* and ‘2′ (*Ht1, Ht2, Ht3* and *HtN1*) differential lines were obtained after the molecular screening of 120 genotypes with *Ht* resistant gene-specific primer (unpublished). This methodology was used for the identification of physiological races of *E. turcicum*.

To assess the conformability of avirulent response, the frequency of plant *Ht* resistance responses for each *Ht* genotype (number of plants displaying resistant phenotype: two differential plant replicates) was calculated. Data on isolate virulence and race were compiled using the Habgood–Gilmour spreadsheet (HiGiS) for each district separately and collectively [14]. Districts’ race distribution, isolate virulence complexity (the amount of *Ht* resistance genes with which an isolate has a susceptible interaction), isolate virulence frequency distributions, and generally used simple diversity index were calculated and presented.

### 2.3. Screening of Isolates in Portrays

Plastic portrays 38 × 24 × 8 cm in diameter were filled with a sterilized mixture of compost + soil + sand (3:3:1 ratio). In portrays, three seeds of each genotype were sowed. The day/night temperature was kept at 26.2 °C/20.2 °C throughout the experiment, and the plants were exposed to diffused sunlight throughout the day. Depending on their needs, plants were irrigated. Genotype ‘6′ was used as a positive control to confirm that the inoculation and disease development were successful. If all three seedlings appeared at once, one plant from each genotype was taken out, and the two surviving plants were used as two replicates for each *E. turcicum* isolate. A single *E. turcicum* isolate to be described was suspended and injected into two plants of each genotype. At two temperatures, 26/22 °C day/night and 22/22 °C day/night, with light intensities ranging from 35 to 50 K Lux, early research evaluated the development of NCLB disease [15].

### 2.4. Preparation of Inoculum and Pathogen Load

The isolates that would be investigated were produced by replicating the first production of single conidial cultures of *E. turcicum* that were stored on PDA at 4 °C in a refrigerator. Cultures were cultivated on PDA at 26 °C for 10 to 12 days with 12-h light/dark cycles in order to induce sporulation. Using a sterile glass slide and sterile water, the conidia were then removed from the surface. The suspension was diluted to 1×10^5^ conidia/mL, conidia were counted using a hemocytometer, and it was then filtered through two layers of sterile muslin cloth.

### 2.5. Disease Development

14 days after inoculation, plants were examined for symptoms of infection. Across all maize lines, virulent *E. turcicum* isolates caused susceptible host reactions that started as gray-green elliptical lesions in the first week, transformed to necrotic lesions in the second week, and grew larger in the third week. Data were analyzed with the software Excel Microsoft 2003 and OPSTAT [16].

## 3. Results

### 3.1. Collection of E. turcicum Isolates

A total of 45 isolates, 9 from Bhagalpur, 9 from Begusarai, 9 from Khagaria, 9 from Katihar and 9 from Samastipur were obtained from the leaf samples collected from 135 fields visited across Bihar in 2020–2021 and 2021–2022. These isolates produced small, pinhead-sized chlorotic lesions 48 to 72 h after inoculation on susceptible maize plants. Clear resistant and susceptible reactions became apparent 10 to 12 days after inoculation.

### 3.2. Race Determination 

Based on the infection reaction (resistant or susceptible) of the isolates on the five differentials 6*Ht*, 33*Ht1*, 15*Ht3*, 55*Ht23N* and 2*Ht123N,* 45 isolates were categorized into five physiological races (0, 1, 3, 23N and 123N). Resistance responses were observed on all the differential plants with *Ht* resistance genes on inoculation with a virulent and avirulent isolate. *E. turcicum* race 3 was the most prevalent race followed by race 0 and race 1 and the least prevalent race was race 123N. Virulence of isolates was found in different reactions due to the presence of the *Ht1, Ht2, Ht3* and/or *HtN1* resistance genes. However, resistance responses of isolates on differential lines having *Ht2, Ht3* and *HtN1* genes were characterized by prominent chlorosis around elongated necrotic lesions of average size 6.97 mm. In the differential line containing *Ht3* genes, the resistance response of isolates was characterized by fewer lesions size of 4.19 mm followed by differential line containing *Ht1* genes with lesions size 3.75 mm. However, in the differential line containing *Ht1, Ht2, Ht3* and *HtN1* genes, the resistance response of isolates was characterized by the least lesion size of 2.64 mm. The smaller lesions were recorded in the avirulent races characterized as race 0 with lesion length size less than 1.00 mm (Table 2).

On the basis of lesion length, all the isolates were divided into three groups, virulent, moderately virulent and less virulent. The six isolates KhEt4, KhEt6, KhEt7, KhEt9, KaEt3 and KaEt7 come under highly virulent isolates with an average lesion length size of 6.97 mm characterized as race 23N (Figure 1). The twelve isolates BhEt3, BhEt4, BhEt6, BhEt7, BhEt8, BeEt4, BeEt9, KhEt5, KaEt1, KaEt2 KaEt5 and SaEt7 come under moderately virulent isolates with average lesion length size 4.19 mm characterized as race 3 (Figure 2), the ten isolates BhEt9, BeEt1, BeEt6, BeEt7, KhEt1, KhEt8, KaEt9, SaEt5, SaEt6 and SaEt9 also come under moderately virulent isolates with average lesion length size of 3.75 mm were characterized as race 1 (Figure 3), the six isolates BhEt2, BeEt3, KhEt3, SaEt1, SaEt3 and SaEt8 come under least virulent isolates with average lesion length size 2.64 mm were characterized as race 123N (Figure 4) and the eleven isolates BhEt1, BhEt5, BeEt2, BeEt5, BeEt8, KhEt2, KaEt4, KaEt6, KaEt8, SaEt2 and SaEt4 did not produce any lesion and it was characterized as avirulent race 0 with average lesion size below 1.00 mm (Figure 5).

All five races were present in each of the five districts (Bhagalpur, Begusarai, Khagaria, Katihar and Samastipur) of Bihar except race 23N was absent in Bhagalpur, Begusarai and Samastipur whereas, race 123N was absent in Katihar district. Race 3, race 0, and race 1 were the most commonly observed races with 24.4%, 22.2% and 26.6% frequency of the total isolates tested, respectively. Race 3 was the predominant race in Bhagalpur and Katihar, whereas race 0 was predominant in Begusarai and Katihar and race 1 was predominant in Begusarai and Samastipur. Race 23N was predominant in Khagaria and race123N was predominant in Samastipur (Table 3).

The prevalence of all five races varies in different agro-climatic zones such as in zone I presence maximum of race 1 followed by race 0, race 123N, race 3 whereas, race 23N was found absent. In zone II maximum prevalence of race 23N was followed by race 0 or race 3, race 1 and race 123N. In agro-climatic zone IIIB, the prevalence of race 3 was maximum followed by race 0, race 1 or race 123N whereas, race 23N was found absent. 

The frequency of virulence to the specific *Ht* genes was also examined across districts (Table 4). Of the isolates tested, 24.4% were avirulent to all *Ht* genes, 22.2% were virulent to *Ht1*, 26.6% were virulent to *Ht3,* and 13.3% were virulent to *23N* and *123N* for each gene.

## 4. Discussion

In the study, 45 isolates of *E. turcicum* were isolated from five different maize-growing districts representing three agro-climatic zones of Bihar. Among all isolates, five races were identified on the basis of disease reactions on differential lines. Race 23N, the highly virulent race are present in the agro-climatic zone II. Moderately virulent races, race 1 and race 3 and avirulent race 0 are evenly distributed over all maize-growing agro-climatic zones of Bihar. Race 123N, the least virulent isolates are also present in all agro-climatic zones of Bihar. In earlier studies, similar results were observed in Bihar with race 1, race 2, and race 3 [10]. Due to the fact that earlier research concentrated on isolates obtained, it is plausible that more races have evolved over time in India [10].

Incompatible interactions were induced by inoculating race 0 on near-isogenic lines 33*Ht1*, 15*Ht3*, 55*Ht23N* and 2*Ht123N*, whereas the compatible interaction was studied by inoculating the same race 0 isolate on 6*Ht* without resistance genes. Plants with no resistance genes are the only ones affected by race 0. On the other hand, plants expressing the resistance genes *Ht2, Ht3*, and *HtN1* are vulnerable to race 23N isolates [17]. A succession of the biotrophic and necrotrophic processes of infection defines *E. turcicum* as a hemibiotroph. Penetration and colonization of the xylem are features of the biotrophic process. The pathogen showed evidence of having the ability to enter the xylem vessels even during the incompatible interaction. However, in the compatible interaction, hyphae developed and disseminated into the cells of the vascular bundles [18]. The resistance manifested at the time point of xylem colonization was critical for subsequent phases in the pathogenesis in all *Ht*-resistant lines evaluated in this work (*Ht1, Ht2, Ht3*, and *HtN1*).

Each resistant line’s unique symptom expression and fungal colonization patterns clearly suggest that each *Ht* resistance gene codes for a different type of resistance mechanism. Early infection phases in the differential line 33*Ht1* show large necrotic lesions encircled by chlorosis for the *Ht1* resistance. Resistance in this situation manifests as chlorosis, smaller lesions, or fewer lesions overall. The *Ht3* gene is easily recognized in the differential line 15*Ht3* by chlorotic patches with medium-sized lesions. *Tripsacum floridanum* [7], which is not an alternate host for *E. turcicum*, provided the source of the *Ht3* gene. This suggests that a mechanism resembling non-host resistance may be responsible for the longevity of the *Ht3* resistance trait. As an exception, the resistance conferred by *HtN1* is characterized as quantitative resistance [19].

These findings are consistent with recent research that found a minimal prevalence of race 23N and race 123N and the highest prevalence of race 1, race 2 and race 0 throughout Bihar. Because virulence appears to be impacted by a single gene, it is confounding when one of these genes is present but not the other [8]. Despite this, numerous investigations have revealed races that are virulent to one gene but not the other [20,21,22,23].

In all districts, races that were *Ht1*, *Ht3,* or both were fairly common. Given that the *HtN1* gene may have earlier been used in breeding programmes and that virulence has been documented in the United States. *E. turcicum* populations [9,24,25], pathogenicity to *HtN1* was expected. The fewest races exhibiting virulence to both *Ht23N* and *Ht123N* were found among the examined isolates. In all districts, virulence to these *Ht* genes was noted. Despite virulence being present in the majority of districts, virulent isolates were the least common within districts. These findings are consistent with those of other research that found a low prevalence of *Ht2* and *Ht3* pathogenicity throughout the United States [9,21,26,27]. The presence of virulence in one of these genes without the presence of the other is confusing because virulence appears to be conferred by the same single gene [28]. In spite of this, multiple studies have reported races virulent to one gene without virulence to the other gene [20,21,22,23]. However, it is unclear what contributes to the variation in virulent races. Because nearly all of the commercial corn production in Bihar is with hybrids developed by private companies, the degree of *Ht* gene deployment is publicly unknown, which makes interpretation of the results more difficult.

## 5. Conclusions

*E. turcicum* race 3 was the most prevalent race having 26.6% frequency followed by race 0 (24.4%) and race 1 (22.2%) and the least prevalent races were race 23N and 123N having 13.3% each. Race populations were diverse within agro-climatic zones as well as districts. Diverse resistance response of different isolates was observed on differential lines having different resistant genes. Isolates on differential lines having *Ht2, Ht3* and *HtN1* genes were characterized by prominent chlorosis with maximum necrotic lesions of size 6.97 mm. The differential line containing *Ht3* genes produced second larger lesions of size 4.19 mm followed by the differential line containing *Ht1* genes, causing lesions of size 3.75 mm and the differential line containing *Ht1, Ht2, Ht3* and *HtN1* genes causing a lesion of size 2.64 mm. The smallest lesions were recorded in the avirulent races characterized as race 0 with lesion length size less than 1.00 mm. The *Ht1* gene may not be able to effectively control NCLB disease in most locations due to its wide pathogenicity, but several other *Ht* genes may. The *Ht2* and *Ht3* genes would typically offer the strongest protection against *E. turcicum* populations. The breeding programme might benefit from using more *Ht* genes. There was a considerable range of racial groups observed throughout time periods and places. They also concluded that *E. turcicum* inoculum may travel across long distances, which may lead to the emergence of new races and ethnic diversity in various professional location. As race complexity (virulence to multiple *Ht* genes) grew and frequency decreased, virulence genes may have fitness consequences. This collection’s diversity of race groups does suggest that its population is genetically diverse, maybe as a result of sexual recombination or mutation as well as some form of selection. *Ht* genes may offer some amount of disease protection when quantitative resistance is strengthened and introgressed into well-known maize lines or when combined with quantitative resistant characteristics, yet it is impossible to say whether or not they will help with better long-term control of NCLB.

## Figures and Tables

**Figure 1 bioengineering-10-00007-f001:**
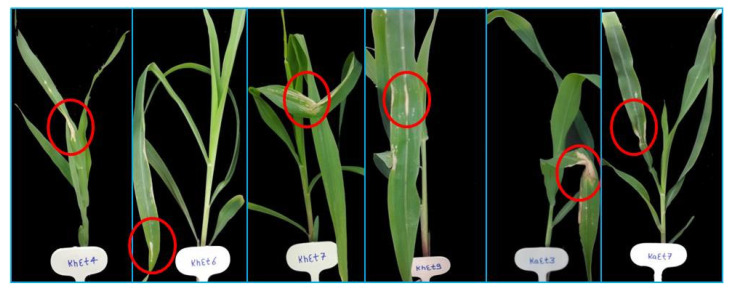
Disease reaction of *E. turcicum* race 23N on differential line ‘55′ after 14th days of inoculation.

**Figure 2 bioengineering-10-00007-f002:**
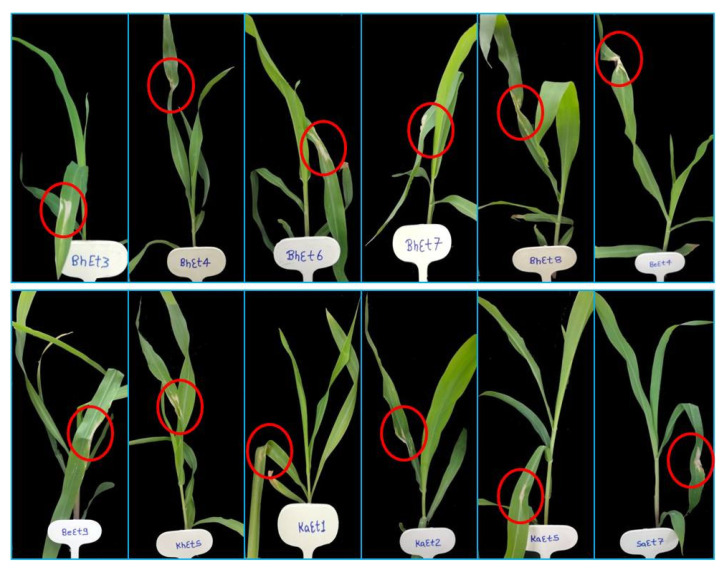
Disease reaction of *E. turcicum* race 3 on differential line ‘15′ after 14th days of inoculation.

**Figure 3 bioengineering-10-00007-f003:**
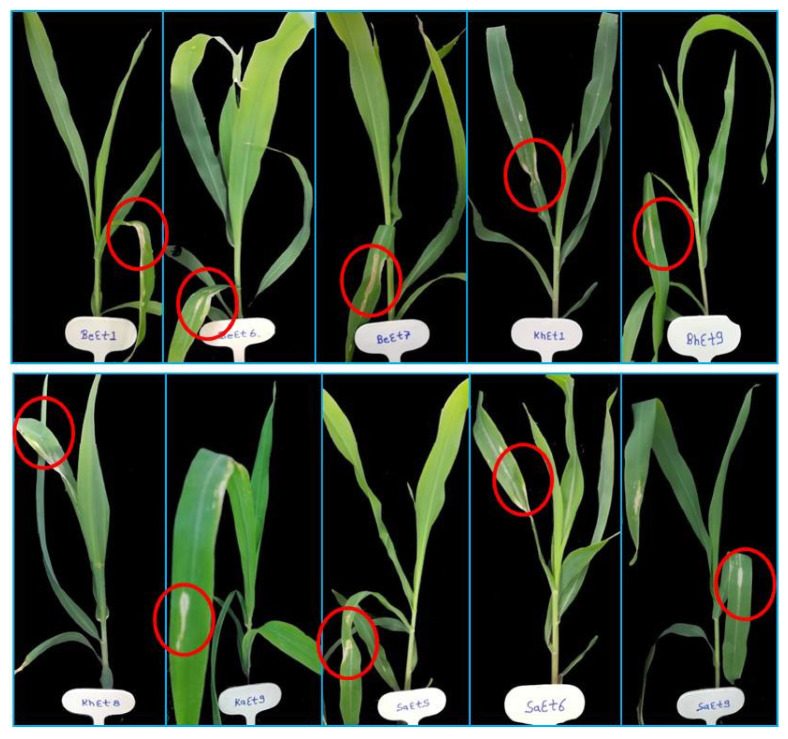
Disease reaction of *E. turcicum* race 1 on differential line ‘33′ after 14th days of inoculation.

**Figure 4 bioengineering-10-00007-f004:**
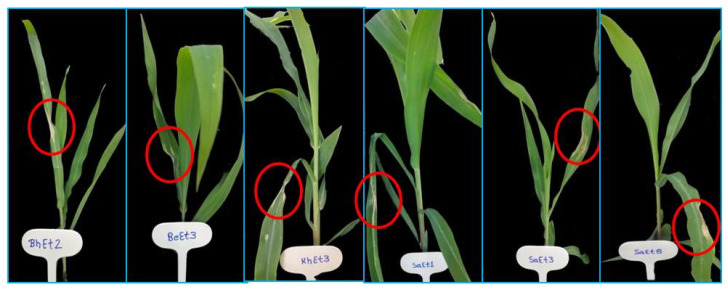
Disease reaction of *E. turcicum* race 123N on differential line ‘2′ after 14th days of inoculation.

**Figure 5 bioengineering-10-00007-f005:**
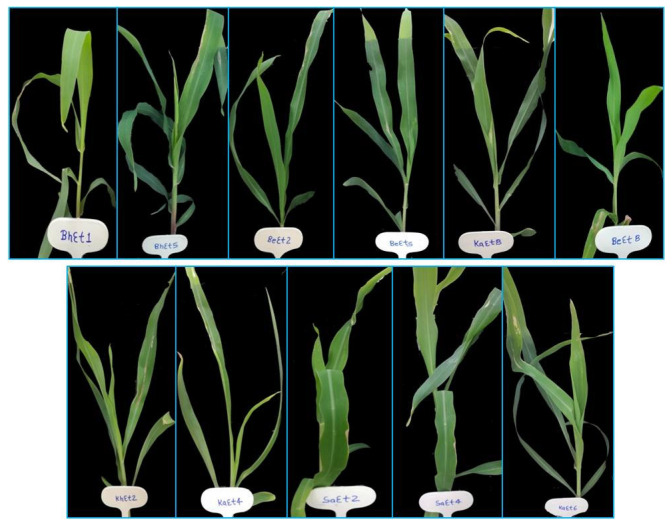
Disease reaction of *E. turcicum* race 0 on differential line ‘6′ after 14th days of inoculation.

**Table 1 bioengineering-10-00007-t001:** Details of the places surveyed for collection of Northern corn leaf blight samples during 2020–22.

S.N.	District	Block	Village	Designated Ioslates
1.	Bhagalpur	Sabour	Sabour	BhEt1
2.			Rajpur	BhEt2
3.			Shankarpur	BhEt3
4.		Naugachhia	Khagra	BhEt4
5.			Pakara	BhEt5
6.			Tatari	BhEt6
7.		Nathnagar	Sahibganj	BhEt7
8.			Rannuchak	BhEt8
9.			Noorpur	BhEt9
10.	Begusarai	Sahabpur Kamal	Parora	BeEt1
11.			Sanha	BeEt2
12.			Chaukiroad	BeEt3
13.		Ballia	Makaspur	BeEt4
14.			Hazipur	BeEt5
15.			Dhabouli	BeEt6
16.		Begusarai sadar	Bahdarpur	BeEt7
17.			Suja	BeEt8
18			Rajaura	BeEt9
19.	Khagaria	Khagaria	Kothia	KhEt1
20.			Durgapur	KhEt2
21.			Ranko	KhEt3
22.		Mansi	Tatha	KhEt4
23.			Mansi	KhEt5
24.			Bakhtiyarpur	KhEt6
25.		Gogri	Maheshkhunt	KhEt7
26.			Jamalpur	KhEt8
27.			Imadpur	KhEt9
28.	Katihar	Kursela	Muradpur	KaEt1
29.			Balthi	KaEt2
30.			Tingharia	KaEt3
31.		Sameli	Chandpur	KaEt4
32.			Khonta	KaEt5
33.			Chhohar	KaEt6
34.		Falka	Pothia	KaEt7
35.			Simaria	KaEt8
36.			Govindpur	KaEt9
37.	Samastipur	Pusa	Pusa	SaEt1
38.			Mahmada	SaEt2
39.			Harpur	SaEt3
40.		Saidpur	Baswari	SaEt4
41.			Sahuri	SaEt5
42.			Malinagar	SaEt6
43.		Kalyanpur	Shivnagar	SaEt7
44.			Lacchhrampur	SaEt8
45.			Somnaha	SaEt9

**Table 2 bioengineering-10-00007-t002:** Race determination based on differential lines and lesion length (mm) of *E. turcicum* isolates.

Si. No.	Resistant Gene Present	Name of Differential Lines Used	Isolates	Lesion Length (mm)	Race	Avg. Lesion Size (mm)	Pathogen Categorized *
1	Without *Ht* gene	6	BhEt1	1.00	0	0.99	Avirulent
2			BhEt5	1.17			
3			BeEt2	1.00			
4			BeEt5	0.67			
5			BeEt8	1.00			
6			KhEt2	0.83			
7			KaEt4	1.17			
8			KaEt6	1.17			
9			KaEt8	1.00			
10			SaEt2	0.83			
11			SaEt4	1.00			
12	*Ht1*	33	BhEt9	3.67	1	3.75	Moderately Virulent
13			BeEt1	3.67			
14			BeEt6	3.50			
15			BeEt7	3.67			
16			KhEt1	4.00			
17			KhEt8	3.83			
18			KaEt9	3.67			
19			SaEt5	3.83			
20			SaEt6	3.83			
21			SaEt9	3.83			
22	*Ht3*	15	BhEt3	4.67	3	4.19	Moderately Virulent
23			BhEt4	4.33			
24			BhEt6	4.33			
25			BhEt7	3.67			
26			BhEt8	4.67			
27			BeEt4	4.33			
28			BeEt9	4.00			
29			KhEt5	4.33			
30			KaEt1	3.83			
31			KaEt2	4.00			
32			KaEt5	4.33			
33			SaEt7	3.83			
34	*Ht2, Ht3* and *HtN1*	55	KhEt4	7.17	23N	6.97	Highly virulent
35			KhEt6	7.17			
36			KhEt7	7.83			
37			KhEt9	5.83			
38			KaEt3	8.17			
39			KaEt7	5.67			
40	*Ht1, Ht2, Ht3* and *HtN1*	2	BhEt2	2.83	123N	2.64	Less virulent
41			BeEt3	2.67			
42			KhEt3	2.67			
43			SaEt1	2.67			
44			SaEt3	2.50			
45			SaEt8	2.50			
			L.S.D. (1%)	0.44			
			C.V.	7.83			

* The pathogen was categorized based on lesion length (mm) at 14 days after inoculation.

**Table 3 bioengineering-10-00007-t003:** Frequency of *E. turcicum* isolates of each race found in different districts.

	Percent *E. turcicum* Race Distribution in Districts					
Race	Bhagalpur	Begusarai	Khagaria	Katihar	Samastipur	Isolates Per Race (%)
0	22.2	33.3	11.1	33.3	22.2	24.4
1	11.1	33.3	22.2	11.1	33.3	22.2
3	55.5	22.2	11.1	33.3	11.1	26.6
23N	0	0	44.1	22.2	0	13.3
123N	11.2	11.1	11.1	0	33.3	13.3
**Total no. of isolates ^a^**	**9**	**9**	**9**	**9**	**9**	**45 ^b^**

^a^ The total number of isolates evaluated from each district. ^b^ The total number of isolates evaluated for race.

**Table 4 bioengineering-10-00007-t004:** Frequency of *E. turcicum* isolates virulent to the *Ht* resistance genes within and across districts.

Districts	Virulent Isolates for Each *Ht* Gene (%) ^a^					
	Avirulent	*Ht1*	*Ht3*	*Ht23N*	*Ht123N*	No. of Isolates Per Districts
Bhagalpur	18.1	10.0	41.6	0	16.6	9
Begusarai	27.2	30.0	16.6	0	16.6	9
Khagaria	9.0	20.0	8.3	66.6	16.6	9
Katihar	27.2	10.0	25.0	33.3	0	9
Samastipur	18.1	30.0	8.3	0	50.0	9
**All districts**	**11**	**10**	**12**	**6**	**6**	**45 ^b^**

^a^ Percent frequencies are equal to the number of isolates virulent to the *Ht* resistance gene divided by the number of isolates collected in each districts. ^b^ The total number of isolates evaluated for virulence to *Ht* resistance genes across districts.

## Data Availability

The original data were available from the corresponding author upon an appropriate request.

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
