# Peer review of "Exserohilum turcicum (Passerini) Leonard and Suggs: Race Population Distribution in Bihar, India"

_bioengineering, 2022, doi:10.3390/bioengineering10010007_

Round 1
Reviewer 1 Report
The presented work has significant scientific and practical value. At the same time, some shortcomings can be noted that reduce it.
In the Discussion section, it is indicated that the prevalence of isolates of the studied pathogen in different agro-climatic zones of the districts differs, but the features of these zones, which, according to the authors, were favorable for the development of specific races of the pathogen, are not indicated.
The article indicates the use of statistical programs, but the results of statistical data processing are not presented in the article.
The article does not provide a description of the studied maize lines and the influence of their genetic and morphological features on the possibility of damage and spread of this disease.
The causative agent (Exserohilum turcicum) affects not only castings, but also ground and underground nodes and other vegetative organs, as well as grains on the cob, however, only leaf damage was taken into account when conducting research.
Author Response
Revered Reviewer,
We would like to thank reviewer for valuable suggestions, insightful comments and inputs have considerably improved the manuscript. Last, but not the least, we thank you once again for considering our manuscript for publication in your esteemed journal. Please find our point-by-point response (in red) to reviewer comments. Further, we have made the necessary changes in the manuscript (in track changes version of revised manuscript).
We are looking forward to hearing from you.
Thanks and regards
Md. Arshad Anwer

Reviewer 2 Report
Please check the comments in the pdf files, mostly regarding the references style.
I also would suggest to check the layout of the paper: table 2 is split over two pages and this is not easy to read.

Author Response

(The authors gave the same response as above.)

Reviewer 3 Report
The manuscript is quite well presented, drafted and analyzed. Data generated is novel and statistical analysis is robust and appropriate. The authors have done a very good job in analyzing and explaining the data generated in sections and sub sections with appropriate and well clarified graphics. The language at parts is a bit monotonous. It will be good if the authors do a through revision of the English language and use smaller sentences instead of using long complex sentences to make it much easier for readers to follow. Moderate revision of the language and punctuations will certainly make the manuscript more effective in reaching the target audience successfully and get better cited. This is an important contribution in the respective field of agriculture research and is recommended for publication with sone minor language revisions.
Author Response

(The authors gave the same response as above.)
